# Integrating National Ecological Observatory Network (NEON) Airborne Remote Sensing and In-Situ Data for Optimal Tree Species Classification

**Victoria M. Scholl** [1,2,*] **, Megan E. Cattau** [1,3] **, Maxwell B. Joseph** [1] **and Jennifer K. Balch** [1,2]

1   Earth Lab, University of Colorado Boulder, 4001 Discovery Drive, Suite S348 611 UCB,
    Boulder, CO 80303 USA; megancattau@boisestate.edu (M.E.C.); maxwell.b.joseph@colorado.edu (M.B.J.);
    jennifer.balch@colorado.edu (J.K.B.)
2   Department of Geography, University of Colorado Boulder, GUGG 110, 260 UCB, Boulder, CO 80309, USA
3   Human-Environment Systems, ERB 4153, 1215 W University Drive, Boise State University,
    Boise, ID 83706, USA
*   Correspondence: victoria.scholl@colorado.edu

**Abstract:** Accurately mapping tree species composition and diversity is a critical step towards spatially explicit and species-specific ecological understanding. The National Ecological Observatory Network (NEON) is a valuable source of open ecological data across the United States. Freely available NEON data include in-situ measurements of individual trees, including stem locations, species, and crown diameter, along with the NEON Airborne Observation Platform (AOP) airborne remote sensing imagery, including hyperspectral, multispectral, and light detection and ranging (LiDAR) data products. An important aspect of predicting species using remote sensing data is creating high-quality training sets for optimal classification purposes. Ultimately, manually creating training data is an expensive and time-consuming task that relies on human analyst decisions and may require external data sets or information. We combine in-situ and airborne remote sensing NEON data to evaluate the impact of automated training set preparation and a novel data preprocessing workflow on classifying the four dominant subalpine coniferous tree species at the Niwot Ridge Mountain Research Station forested NEON site in Colorado, USA. We trained pixel-based Random Forest (RF) machine learning models using a series of training data sets along with remote sensing raster data as descriptive features. The highest classification accuracies, 69% and 60% based on internal RF error assessment and an independent validation set, respectively, were obtained using circular tree crown polygons created with half the maximum crown diameter per tree. LiDAR-derived data products were the most important features for species classification, followed by vegetation indices. This work contributes to the open development of well-labeled training data sets for forest composition mapping using openly available NEON data without requiring external data collection, manual delineation steps, or site-specific parameters.

**Keywords:** airborne remote sensing; tree species classification; National Ecological Observatory Network; machine learning; hyperspectral; multispectral; LiDAR; training data preparation

## 1. Introduction

Mapping and monitoring the tree species composition of Earth's forests using remote sensing technologies is a challenging task that has motivated an active research community over the past three decades [1–7]. In between the global coverage and long-standing historical records of satellite platforms [8–10] and the flexibility and high spatial resolution achieved by unmanned aircraft systems [11–13], airborne platforms provide a valuable intermediate scale of local to regional ecosystem

monitoring capabilities [3,14–17]. Passive optical (multispectral [10] and hyperspectral [18]) and active (such as Light Detection and Ranging (LiDAR) [19]) remote sensing systems have each been used to map tree species on their own with varying levels of success, but combining spectral and structural remote sensing data has been shown to generally improve tree species detection abilities compared to using any of them alone [2,3,14,20–22]. For instance, the combination of hyperspectral and LiDAR data has the ability to differentiate between species with similar reflectance properties but different mean heights [3].

These remotely sensed spectral and structural characteristics of trees are used to predict species using a variety of pixel-based and object-based classification approaches [6,23–25]. Three commonly used classification approaches are non-parametric machine learning techniques including Support Vector Machines (SVM), Random Forest (RF), and neural networks [6,26–28]. SVM and RF tend to perform similarly in terms of classification accuracy and training time [29–31]. Neural networks are increasingly used in ecological remote sensing studies for their ability to identify trends and patterns from data, model complex relationships, accept a wide variety of input predictor data, and produce high accuracies, at the expense of requiring large amounts of training data [13,32–34]. Tree species classification accuracies reported throughout the literature vary widely from approximately 60% to 95%, along with the type and number of sensors used, biodiversity within forests, and classification methods utilized [6]. Many studies highlight the value of deriving metrics such as texture to quantify crown-internal shadows and foliage properties [35], calculating vegetation indices and employing dimensionality reduction when working with hyperspectral data [36], and removing dark or non-green pixel outliers that may contain shadows or soil [37] when preparing remote sensing image features for species classification [6]. Other factors that impact tree species classification accuracy include tree species complexity [38], and the time of acquisition within a year or season, especially for deciduous trees with distinctive phenological patterns [27].

Large amounts of open ecological and remote sensing data are becoming increasingly available in recent years to enable and motivate advancements in tree species classification efforts [6,39]. A notable source of these data is the National Ecological Observatory Network (NEON), a National Science Foundation (NSF) funded Grand Challenge project awarded with the purpose of measuring ecological change for a span of 30 years [40]. NEON provides publicly available data at 81 field sites in 20 distinct eco-climatic domains across the continental United States, Alaska, Hawaii and Puerto Rico [41]. NEON collects airborne remote sensing observations at a subset of these sites every year from the Airborne Observation Platform (AOP). The AOP sensor suite includes multispectral, hyperspectral, and LiDAR instruments to map regional land cover at high spatial resolutions, ranging from 0.1 to 1 m. [42]. NEON generates a series of publicly available image data products from the AOP data and provides documents with detailed descriptions of their standardized data collection protocols and processing algorithms. Within the extent of AOP coverage, NEON field technicians collect terrestrial field-based plant measurements towards the aim of monitoring changes in biodiversity, species abundance, and productivity [41]. In-situ measurements of individual trees are collected at stratified random plot locations at vegetated NEON sites every 1–3 years, including stem locations, species, and crown diameter [43].

NEON data has enabled a new wave of tree detection and classification research, in addition to a need for integrative and reproducible analysis and synthesis [34,37,44,45]. This research has been further accelerated by an ecological data science competition that has tasked research groups with tree crown segmentation, alignment of data, and species classification at the open canopy longleaf pine ecosystem at the Ordway-Swisher Biological Station in Florida [37,46]. A second iteration of this competition is currently underway with the focus of developing methods that generalize to other NEON sites. Preparing accurate training data that connects field-based ground truth species measurements with remote sensing observations is a critical, yet challenging, aspect of these efforts [37,47]). One recent study proposed a new approach to create tree crown training data using a consumer-grade GPS and tablet to spatially match individual trees measured in the field directly

onto an AOP-derived image [47]. Another recent study leveraged existing unsupervised algorithms to delineate tree crown boundaries based on AOP LiDAR canopy height data, and refined these boundaries using almost 3000 hand-annotated bounding boxes drawn around individual trees using AOP multispectral imagery collected at the San Joaquin Experimental Range (SJER) NEON site in California [28]. These studies are making exciting progress towards creating high-quality, high-volume tree species training data at NEON sites. Ultimately, manually creating training data is an expensive and time-consuming task that introduces human decision making and/or may require external data sets or information beyond what is provided publicly within NEON data products and protocols.

Our work explores training data set preparation using openly available NEON data without requiring external data collection or manual delineation steps. We integrate in-situ vegetation measurements with multisensor AOP remote sensing data at the Niwot Ridge Mountain Research Station (NIWO) subalpine forest NEON site in Colorado to evaluate the impact of our hands-off training data preparation approaches on tree species classification accuracy. We create a series of training data sets by representing individual trees as points and circular polygons with various sizes based on in-situ crown diameter measurements. We propose a preprocessing workflow to remove small, suppressed trees and clip areas of overlap between neighboring tree crown polygons in layered canopies to give preference to taller trees that are more likely to be seen by the airborne remote sensing platform. To assess the impact of training data preparation on species classification accuracy, we train random forest (RF) models to predict tree species using each training set. We also evaluate variable importance to assess the contribution of each AOP remote sensing data product for species classification. This work contributes to the open development of forest composition mapping efforts using NEON data without the need for manual delineation or external data sources. We explored the following objectives:

1. Evaluate which training set preparation approach yields the most accurate tree species classification accuracy. We expected smaller tree polygons would capture more valuable variation in canopy features than using stem location points, and capture less noise and neighboring materials than larger circular polygons.
2. Evaluate the value added of our proposed tree crown polygon clipping workflow, which removes tree crown polygons with small area values and clips overlapping tree crown regions based on associated in-situ tree height measurements.
3. Assess the tree species classification accuracies achievable for the four dominant subalpine conifer species in a region of the Southern Rockies, Colorado, USA using the proposed NEON training data preparation approaches.
4. Determine which NEON AOP imagery-derived features are the most important for predicting tree species to help inform overarching tree species classification efforts. We anticipated the hyperspectral imagery to be the most important compared to RGB or LiDAR-derived features.
5. Contribute open reproducible tools so that the NEON data user community can use and build upon these techniques across diverse vegetated ecosystems.

## 2. Materials and Methods

### 2.1. Study Area and Field Data

The NIWO NEON site is located on the eastern slope of the Colorado Front Range, in the Southern Rockies Domain, 27 km west of Boulder, Colorado and 6 km east of the Colorado Continental Divide (Figure 1). The high-elevation (3000 m) Rocky Mountain ecosystems present here include alpine tundra and subalpine coniferous forests near the treeline dominated by subalpine fir (*Abies lasiocarpa*), lodgepole pine (*Pinus contorta*), Engelmann spruce (*Picea engelmannii*), and limber pine (*Pinus flexilis*).

At partially forested sites such as NIWO, field technicians collect in-situ vegetation measurements from distributed and tower plots every one to three years following the Terrestrial Organismal Sampling protocol for Measurement of Vegetation Structure (described in NEON.DOC.000987).

Distributed plots are measured using the vegetation structure protocol but tower plots are not, because most of the tower airshed is above tree line at NIWO. Field data collected within each 20 m × 20 m sampling plot include the location and species of tree stems (Figure 2), total stem height, and maximum crown diameter. These data are provided in the Woody Plant Vegetation Structure data product (NEON data product ID: DP1.10098.001). An offset mapping technique is used to determine the within-plot location of mapped tree stems relative to permanent plot reference points with precisely known spatial coordinates. A TruPulse 360R laser rangefinder (30 cm accuracy) is used to measure the distance and azimuth of each stem relative to the permanent plot reference point. Crown diameter is measured to the nearest 10 cm using the laser rangefinder.

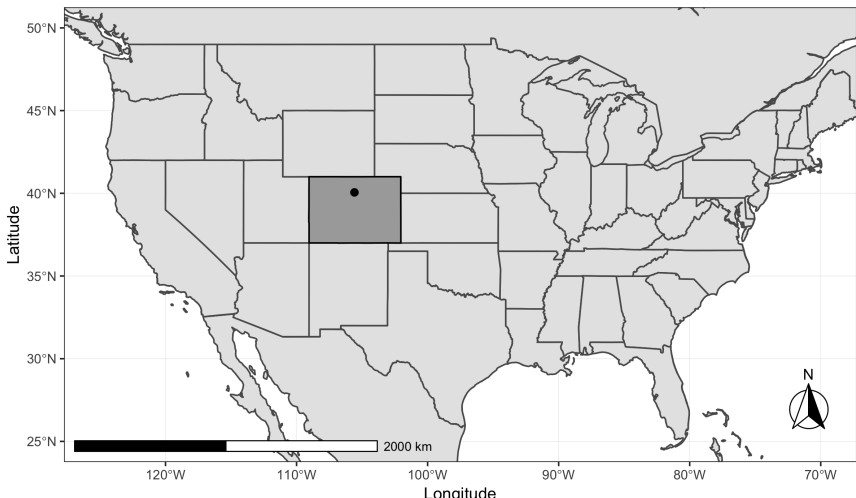

**Figure 1.** Map showing the location of the Niwot Ridge Mountain Research Station (NIWO) site in NEON Domain 13, Southern Rockies and Colorado, in the United States. Latitude: 40.05425, Longitude: −105.58237.

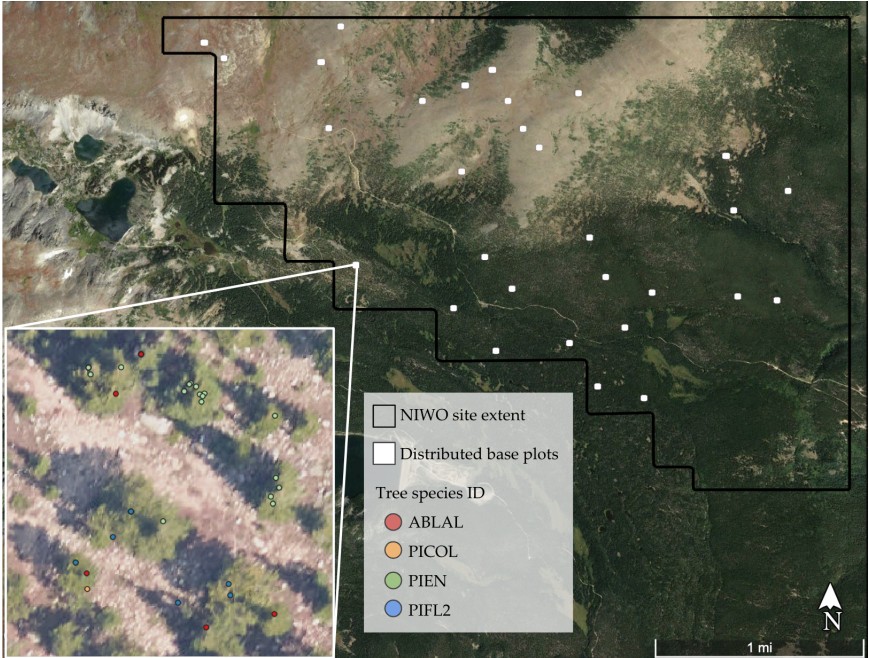

**Figure 2.** Map showing the extent of the NEON NIWO site and distributed sampling plots. Inset zoomed view shows an example of individual tree locations along with their associated species for a single sampling plot.

We downloaded woody vegetation structure data from the NEON Data Portal [48]. The most recent woody plant vegetation structure data available at the NIWO site at this time were collected during late summer/early fall of 2015 and 2016, and consist of 699 individual trees with stem coordinates and crown measurements (Table 1).

**Table 1.** Tree species recorded at the NIWO NEON site for all sampling plots combined in the woody vegetation structure data product.

| Scientific Name (Common Name) | Species Code | Number of Mapped Trees |
|---|---|---|
| *Abies lasiocarpa* (Subalpine fir) | ABLAL | 249 |
| *Pinus contorta* (Lodgepole pine) | PICOL | 112 |
| *Picea engelmannii* (Engelmann spruce) | PIEN | 264 |
| *Pinus flexilis* (Limber pine) | PIFL2 | 74 |

### 2.2. Airborne Remote Sensing Data

NEON's Airborne Observation Platform (AOP) consists of three remote sensing instruments mounted into a DeHavilland DHC-6 Twin Otter aircraft to collect hyperspectral, multispectral, and LiDAR imagery [41]. Coincident capture of these three data facilitates high spatial resolution ecosystem monitoring. During its annual flight campaign, the AOP surveys 75% of core NEON sites on a rotating basis throughout the country. The AOP flight season runs from March to October. NEON terrestrial sites are scheduled to be flown during periods of at least 90% peak vegetation greenness for phenological consistency. The aircraft is flown at an altitude of 1000 m above ground level and a speed of 100 knots to achieve meter-scale hyperspectral and LiDAR raster data products and sub-meter multispectral imagery (described in NEON document NEON.DOC.002236). The data products are publicly available on NEON's data portal as both flight lines and 1 km by 1 km mosaic tiles. When multiple flight lines cover a given tile, the most-nadir pixels are selected for the final mosaic. We utilized the mosaic data products in this study.

The NEON Imaging Spectrometer (NIS) is a pushbroom collection-style instrument that measures reflected light energy with 426 spectral bands, spanning the visible (380 nm) to shortwave infrared (2510 nm) wavelengths in 5 nm increments. The NIS instantaneous field of view (IFOV) is 1 mrad, which yields a ground sample distance (GSD) of 1 m at the reported flight altitude of 1000 m above ground level (described in NEON document NEON.DOC.001290). The NIS-derived raster data products are generated at a 1 m spatial resolution. NEON implements calibration and atmospheric corrections to convert at-sensor radiance to surface reflectance. In addition to 426-band hyperspectral surface reflectance (NEON data product ID: DP3.30006.001), NEON also provides vegetation indices (NEON data product ID: DP3.30026.001), a collection of seven spectral indices that are known to be indicators of plant health: Normalized Difference Vegetation Index (NDVI), Enhanced Vegetation Index (EVI), Atmospherically Resistant Vegetation Index (ARVI), Photochemical Reflectance Index (PRI), Normalized Difference Canopy Lignin Index (NDLI), Normalized Difference Nitrogen Index (NDNI), and Soil-Adjusted Vegetation Index (SAVI). These vegetation indices are calculated using well-known equations from the scientific literature, using NIS surface reflectance data as the inputs.

Discrete and waveform LiDAR were collected using the Optech ALTM Gemini system at a spatial resolution of approximately 1–4 points/waveforms per square meter, using a near-infrared laser wavelength of 1064 nm. Using the discrete return point cloud data, NEON creates Digital Terrain Model (DTM) and Digital Surface Model (DSM) raster data products each with a spatial resolution of 1 m to match that of the NIS imagery. From the DTM, NEON generates Slope and Aspect raster data products to characterize the underlying bare ground terrain surface (NEON data product ID: DP3.30025.001). Slope is determined as the angle between a plane tangential to the local terrain surface and a plane tangential to the local geoid surface, reported in degrees. Aspect is the direction of the steepest slope, given in degrees referenced to grid north. Using the DTM and DSM, NEON also generates the Ecosystem Structure data product, a Canopy Height Model (CHM) raster where each

pixel value quantifies the height of the top of the canopy above the ground (NEON data product ID: DP3.30015.001). We used the Slope, Aspect, and Ecosystem Structure raster data products in our study to capture vegetation structure and topographic characteristics.

The AOP payload also includes an Optech D8900 commercial off-the-shelf digital camera that measures 8-bit light intensity reflected in the red, green, and blue (RGB) visible wavelengths. With a 70 mm focal length lens, the digital camera has a 42.1 deg cross-track field-of-view (FOV) and achieves a GSD of 0.086 m at a flight altitude of 1000 m. The raw RGB images are corrected for color balance and exposure, orthorectified to reduce geometric and topographic distortions and map the RGB imagery to the same geographic projection as the NIS and LiDAR imagery, and ultimately resampled to a spatial resolution of 0.1 m (NEON data product ID: DP3.30010.001), the digital camera imagery can aid in identifying fine features such as boundaries of individual tree crowns in dense canopy that are not as visible in the other airborne data products.

Together, these three AOP instruments capture spectral and structural characteristics of ecosystems present at NEON sites (Figure 3). The closest date of available airborne remote sensing data products to the in-situ vegetation measurements was September 2017 [48].

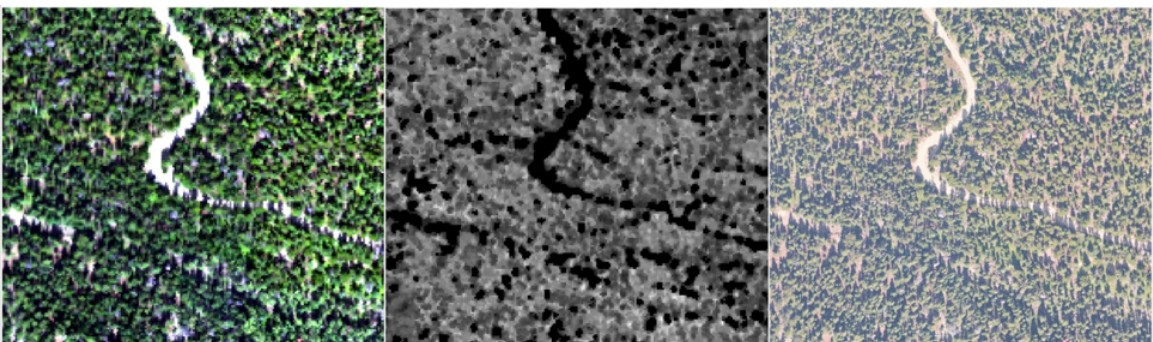

**Figure 3.** Comparison of raster data products from each of the three NEON Airborne Observation Platform (AOP) instruments at the Niwot Ridge Mountain Research Station site: hyperspectral true-color composite with 1 m pixel size using three of the available 426 NEON Imaging Spectrometer (NIS) bands at approximate wavelengths of 450 nm, 555 nm, and 620 nm (**left**), discrete Light Detection And Ranging (LiDAR)-derived canopy height model with 1 m pixel size, where black indicates ground and brighter pixels represent taller canopy height above ground (**center**), and digital camera RGB composite with 10 cm pixel size (**right**).

### 2.3. Proposed Reference Data Preprocessing

The in-situ woody vegetation structure data include tree location and crown diameter measurements for each mapped tree stem within NEON sampling plots. We generated three different "raw" reference data sets to investigate their influence on species classification: (1) tree stem point locations, (2) circular polygons with the maximum crown diameter of each tree, and (3) circular polygons with half of the maximum crown diameter of each tree to serve as an intermediate step between points and large polygons (Figure 4). Many of the points and polygons are overlapping, so that multiple trees will be associated with individual pixel locations. Also, there are duplicate polygons in some locations, which are generated as a result of multi-bole entries present in the reference data. We approximated tree crowns as circles in this study due to the lack of more specific crown shape measurements. In reality, tree crowns at NIWO are often irregularly shaped and asymmetrical due to wind, sun exposure, and proximity of neighboring vegetation.

To investigate the potential value of preprocessing these "raw" reference data, we designed an experimental workflow to remove small polygons and clip overlapping polygons to preserve the taller trees, which are more likely visible in the airborne imagery. In this workflow, we first identified identical multi-bole entries and removed any duplicate points or polygons from each raw data set. We then applied an area threshold (two square meters) to remove small trees. We selected this threshold

with the coarser resolution of the AOP-derived imagery in mind, where one pixel has an area of one square meter. By preserving the trees with larger crowns, we believed that we could extract purer spectra for the training sets as opposed to extracting mixed pixel spectra containing signals from neighboring vegetation and other background materials. "Occluded" tree crown polygons, those which are associated with shorter heights and completely within the boundaries of other tree crown polygons, are also present in the initial reference data sets (Figure 4). Since they likely cannot be observed from the airborne perspective, we identified and removed "occluded" polygons from subsequent analysis. We checked the remaining polygons for overlap with neighboring polygons. Each polygon had an associated height value, as this is one of the measurements collected in the in-situ protocol. For each overlapping pair of polygons, we clipped the shorter polygons. If the remaining clipped area was smaller than the aforementioned area threshold, we removed the polygon from subsequent analysis. After following these steps, we generated a collection of "clipped" polygons to reduce the occurrence of multiple trees being associated with a single pixel in the remote sensing imagery (Figure 5).

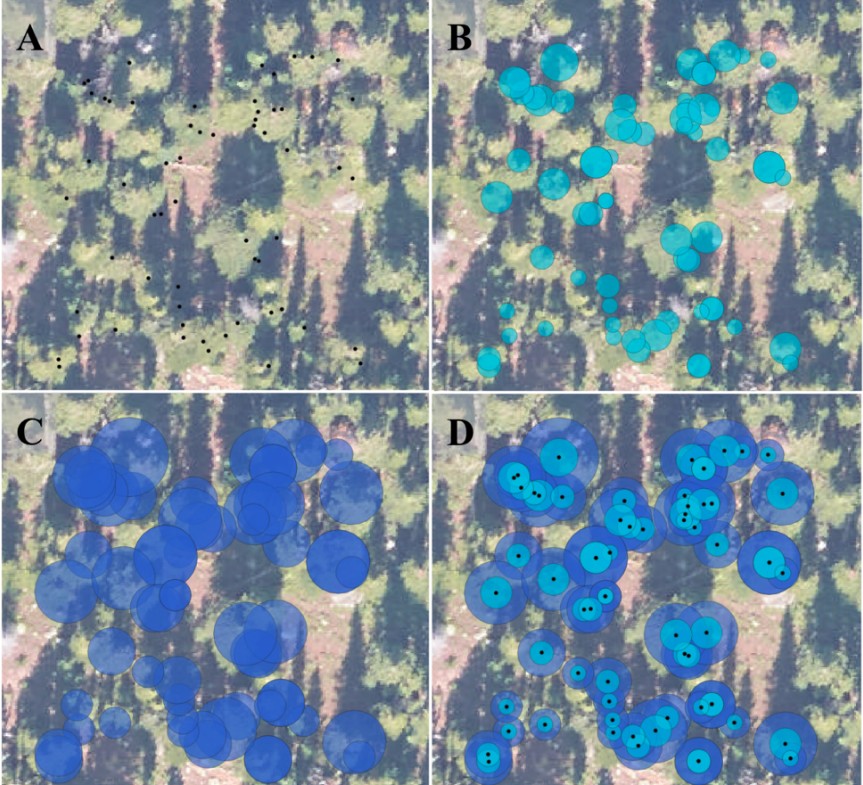

**Figure 4.** "Raw" reference data sets created using all of the input woody vegetation measurements at the NIWO_015 distributed base plot with dimensions of 20 × 20 m, located at 451146 m East, 4432366 m North: (**A**) Tree stem points. (**B**) Circular polygons created with half the maximum crown diameter for each tree. (**C**) Circular polygons created with the maximum crown diameter for each tree. (**D**) All of these points and polygons are displayed together. Polygons are displayed with 50% opacity to help illustrate the areas of overlap between adjacent polygons, as well as the presence of multi-bole entries which appear opaque where multiple identical polygons are present. The base layer is RGB image data collected by the AOP with 10 cm spatial resolution.

We created the following six sets of reference data to evaluate species classification accuracy at the NEON NIWO site in this study: (1) points for all mapped tree stems (Figure 4A), (2) polygons generated using the respective maximum crown diameter for all of the tree stems (Figure 4C), (3) polygons generated using half of the respective maximum crown diameter for all of the tree stems (Figure 4B), (4) "clipped" half-diameter polygons (Figure 5B), (5) "clipped" maximum-diameter

polygons (Figure 5C), and (6) points for mapped tree stems corresponding to the center locations of the clipped half-diameter polygons (Figure 5A).

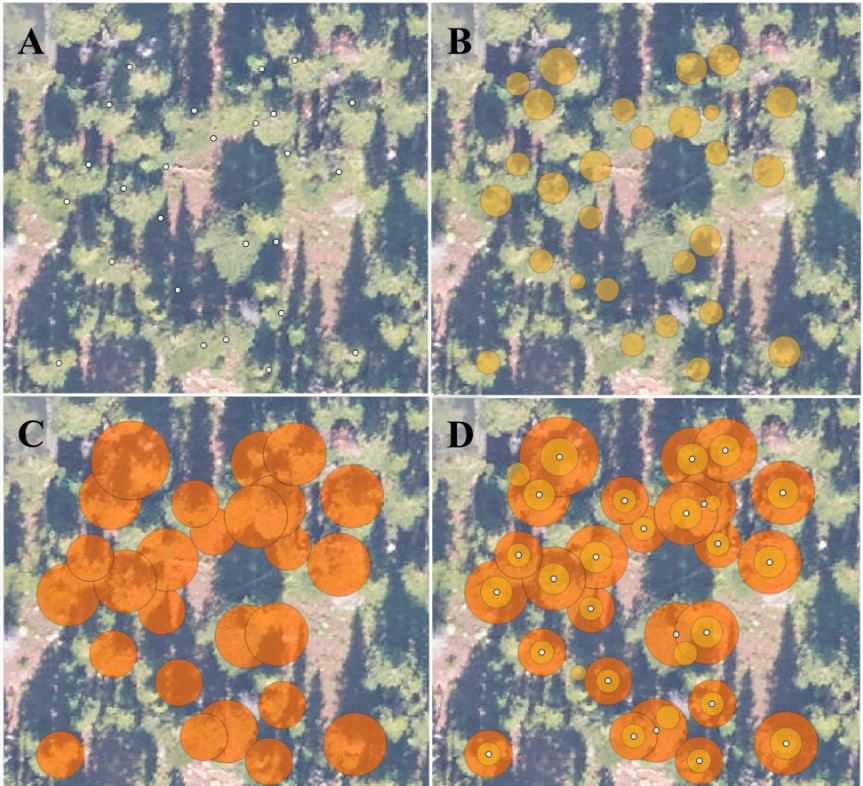

**Figure 5.** "Clipped" reference data sets generated using the proposed preprocessing workflow to filter out small tree crowns and clip overlapping crowns to preserve taller trees at the NIWO_015 distributed based plot: (**A**) Tree stem points corresponding to the clipped half-diameter polygons. (**B**) Clipped half-diameter polygons. (**C**) Clipped maximum-diameter polygons. (**D**) All of these points and polygons are displayed together. Polygons are displayed with 50% opacity to help illustrate the areas of overlap between adjacent polygons. Note that there is no longer overlap between adjacent polygons, and multi-bole entries have been removed. The base layer is RGB image data collected by the AOP with 0.1 m spatial resolution.

### 2.4. Remote Sensing Feature Extraction

To train the tree species classifiers, we utilized NEON AOP-derived images that describe the spectral, structural, and environmental characteristics of vegetation at the NIWO site (Table 2). The NIS captures hyperspectral surface reflectance in 426 bands throughout the visible, near infrared, and shortwave infrared wavelengths. We removed water-vapor absorption bands (1340 to 1445 nm and 1790 to 1955 nm), yielding 372 individual wavelength bands. We used principal component analysis (PCA) to reduce the dimensionality of the hyperspectral data with many highly correlated bands. PCA is a commonly used technique to transform large data sets with correlated dimensions into a new set of orthogonal dimensions or principal components [6,49]. The first two principal components explained a cumulative proportion of 0.96 of the total variance, so we included the first two principal components ("PC1" and "PC2") as features, similarly as [15]. In addition to including the first two principal components, we also included all seven of the Vegetation Indices calculated by NEON as descriptive features since they capture spectral differences between specific wavelengths that relate to vegetation health and chemical composition.

**Table 2.** Remote-sensing-derived features to train random forest models for species classification.

| Feature Name | Description | Inputs / Equations | Reference |
|---|---|---|---|
| PC1, PC2 | 1st and 2nd principal components | 426-band Hyperspectral reflectance (372 after removing bad bands due to atmospheric absorption) 381–2509 nm with 5 nm spacing | [49] |
| Vegetation Indices | | Hyperspectral reflectance bands: | |
| | Normalized Difference Vegetation Index (NDVI) | $NDVI = \dfrac{\lambda_{860} - \lambda_{650}}{\lambda_{860} + \lambda_{650}}$ | [50] |
| | Enhanced Vegetation Index (EVI) | $EVI = \dfrac{2.5 \times (\lambda_{860} - \lambda_{650})}{(\lambda_{860} + (6 \times \lambda_{650}) - (7.5 \times \lambda_{470}) + 1)}$ | [51] |
| | Atmospherically Resistant Vegetation Index (ARVI) | $ARVI = \dfrac{\lambda_{860} - [\lambda_{650} - \gamma(\lambda_{470} - \lambda_{650})]}{\lambda_{860} + [\lambda_{650} - \gamma(\lambda_{470} - \lambda_{650})]}$ | [52] |
| | Canopy Xanthophyll, or Photochemical Reflectance Index (PRI) | $PRI = \dfrac{\lambda_{531} - \lambda_{570}}{\lambda_{531} + \lambda_{570}}$ | [53] |
| | Normalized Difference Lignin Index (NDLI) | $NDLI = \dfrac{log(\frac{1}{\lambda_{1754}}) - log(\frac{1}{\lambda_{1680}})}{log(\frac{1}{\lambda_{1754}}) + log(\frac{1}{\lambda_{1680}})}$ | [54] |
| | Normalized Difference Nitrogen Index (NDNI) | $NDNI = \dfrac{log(\frac{1}{\lambda_{1510}}) - log(\frac{1}{\lambda_{1680}})}{log(\frac{1}{\lambda_{1510}}) + log(\frac{1}{\lambda_{1680}})}$ | [54] |
| | Soil-Adjusted Vegetation Index (SAVI) | $SAVI = \dfrac{(1+L)(\lambda_{850} - \lambda_{650})}{\lambda_{850} + \lambda_{650} + L}$ | [55] |
| CHM | Height of canopy above the ground | LiDAR-derived Digital Surface Model (DSM) – Digital Terrain Model (DTM) with modified data pit filling algorithm | [20] |
| Slope | Steepness of bare earth surface | DTM bare earth elevation ratio: height over distance | [56] |
| Aspect | Compass direction of steepest slope | DTM bare earth elevation degrees clockwise from North | [56] |
| rgb_mean_sd_R rgb_mean_sd_G rgb_mean_sd_B | Mean plus standard deviation of red, green, and blue (RGB) image intensity values | RGB multispectral bands | [35] |

Where $\lambda_n$ is reflectance at the specified wavelength in units of nanometers. $\gamma$ is a weighting constant based on aerosol type and atmospheric compensation for ARVI. *L* is a correction factor to account for different soil conditions for SAVI.

We spatially resampled the multispectral digital camera RGB intensity bands from their initial 10 cm spatial resolution to match the 1 m grid size of the other remote sensing data products. Within each 1 m by 1 m grid cell, we calculated the aggregated mean and standard deviation of RGB intensities. The RGB imagery captures fine-scale canopy variations such as shadows. The mean plus standard deviation for each of the RGB channels were included as features to capture the average spectral intensity along with variation in intensity per pixel. From the LiDAR-derived CHM, we included the height above ground at each pixel location. Although canopy height alone has limited value for species classification due to its dependence on individual tree age [30], it has been found to improve classification accuracies in some studies [20] and is useful for filtering non-canopy pixels.

The slope and aspect products describe the orientation and steepness of the underlying "bare earth" surface, which are important microclimate characteristics that influence species survival within a given environment [56]. For each 1 km by 1 km tile, we combined all of these coincident airborne remote sensing data layers into raster stacks. We excluded any pixels with a height of 0 m in the canopy height model, as we assumed they contain signal from the ground. We then used reference data set to extract features from the remote sensing data stacks by determining which pixels intersect with each of the points or polygons. We extracted spectral reflectance curves from the AOP hyperspectral data for the points and polygons within each reference set to see how spectrally similar each of the tree species appear. For subsequent tree species classification, each AOP pixel was a sample described by the remote sensing-derived spectral and structural features.

*2.5. Random Forest Classification*

Random forests (RF) are a non-parametric supervised classifiers that use ensembles of decision trees, also known as "bootstrap aggregated" or "bagged" decision trees [57]. Each decision tree in the "forest" considers a random subset of features and only has access to a random set of the training data samples (approximately two-thirds of the collected data). The remaining one-third of the data is then used to predict upon using the previously created decision tree. Whereas using a single decision tree is known to lead to overfitting on the training set, building multiple decision trees by iteratively resampling data from the training set reduces the occurrence of overfitting for greater stability and accuracy. Each decision tree within the ensemble "votes" for a final outcome value and the class with the greatest number of votes is assigned in the end. As a form of data-splitting, this Out-Of-Bag (OOB) bootstrap-resampling procedure can provide a reliable estimate of accuracy for unseen data, although a completely independent test set is recommended as the "a gold standard for tree species classification studies" [6]. There are two hyperparameters that may be tuned when constructing an RF classifier—the number of decision trees (ntree) and the number of variables sampled randomly as each split or stage to grow a single tree (mtry). The Mean Decrease in Accuracy (MDA) and Mean Decrease in Gini (MDG) metrics are commonly used to rank RF predictor variable importance [15,20]. Assessing variable importance is a valuable step for downstream feature selection and model interpretation.

We chose to use RF classification because it handles high-dimensionality data well, requires few user-defined parameters, has a low sensitivity to overfitting and the number of input variables, and provides an intuitive derivation of accuracy and variable importance [29]. We used the randomForest R package to classify tree species, providing AOP-derived spectral and structural features as input and the "taxon ID" label as output [58]. We trained six random forest classifiers, each using a different set of reference data. We initially excluded a random 20% of pixels within the half-diameter polygons, and then trained each of the six RF models using the remaining data within each set. Each RF model was then used to predict species of the 20% independent validation set. We set *ntree* to 5000 to ensure that each one of the input samples was used multiple times during training, and left the *mtry* parameter at its default value, the square root of the number of features. We evaluated each classifier's performance based on internal out of bag error estimates, and overall accuracy of the independent validation set predictions. In addition, we created confusion matrices to assess user's and producer's classification accuracies for each of the four species [59]. We calculated Cohen's Kappa coefficient as well, which takes into account the possibility of classification accuracy by chance [60]. We ranked the AOP-derived descriptive features in order of importance based on the Mean Decrease in Accuracy (MDA) and Mean Decrease in Gini (MDG) metrics for the RF models.

As a result of the filtering and clipping operations, the clipped polygon data sets contained less than half of the number of pixels as the initial NEON data sets. To reduce the influence of sample size and sampling bias on the classification results, we reduced the trees within each "raw" data set to include the same tree IDs present in each corresponding clipped set. All code required to reproduce the analysis is available at https://github.com/earthlab/neon-veg.

## 3. Results

The AOP hyperspectral reflectance curves appear very similar across all four tree species, including the characteristic green peak (550 nm) within the visible wavelength region and the steep slope at the edge between the red and near-infrared regions (750 nm) and the shoulder or flattening off into the near-infrared region (800 nm) (Figure 6). To compare reflectance magnitudes in different wavelength regions, we overlaid all four mean spectral reflectance curves with standard deviation shading (Figure 7). The overlaid mean reflectance curves extracted within the clipped half-diameter polygons have very similar shapes across the conifer species but they appear to be biased or separated vertically to varying degrees. The relationship between reflectance magnitude across species curves differs across wavelength regions. Features and biases in reflectance may aid in differentiating between species during classification.

When we compared the overall classification accuracies for each of the RF models trained using each a different training set (Table 3), the clipped half-diameter polygon training set yielded the highest overall accuracy (OA) values of 69.3% and 60.4% for out-of-bag and independent validation evaluations, respectively.

The user's and producer's accuracies for the clipped half-diameter polygon RF model varied greatly across the four species, from 32.6% to 94.9% (Table 4). *Pinus flexilis* (Limber pine) is consistently the most accurately classified species, while *Abies lasiocarpa* (Subalpine fir) and *Picea engelmannii* (Engelmann spruce) were less accurately classified. *Abies lasiocarpa* is often incorrectly predicted as *Picea engelmannii* (Engelmann spruce). The confusion matrices of the other five training data sets demonstrate analogous species-specific classification performance.

When we ranked the AOP-derived descriptive features in order of importance (Figure 8), the structural features derived from the LiDAR data (aspect, slope, and canopy height) were ranked as the top three most important variables. Variable importance appeared to taper off after this, with the following multispectral- and hyperspectral-derived features listed as the next most important across the MDA and MDG metrics: blue intensity from the digital camera, ARVI (Atmospherically Resistant Vegetation Index), PRI (Photochemical Refelectance Index), and NDVI (Normalized Difference Vegetation Index).

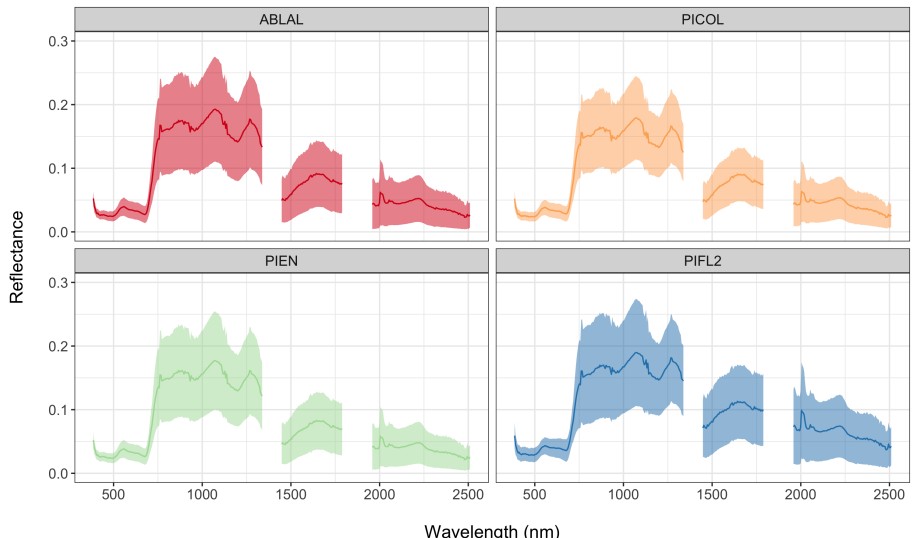

**Figure 6.** Mean hyperspectral reflectance per species from 380 to 2510 nm, extracted from all polygons with half the max crown diameter at the NEON NIWO site for each of the dominant tree species: ABLAL (Subalpine fir), PICOL (Lodgepole pine), PIEN (Engelmann spruce), and PIFL2 (Limber pine). Shading illustrates +/− one standard deviation in reflectance per wavelength. Gaps in the spectra at approximately 1350 nm and 1800 nm are where "bad bands" were removed where there is high atmospheric absorption.

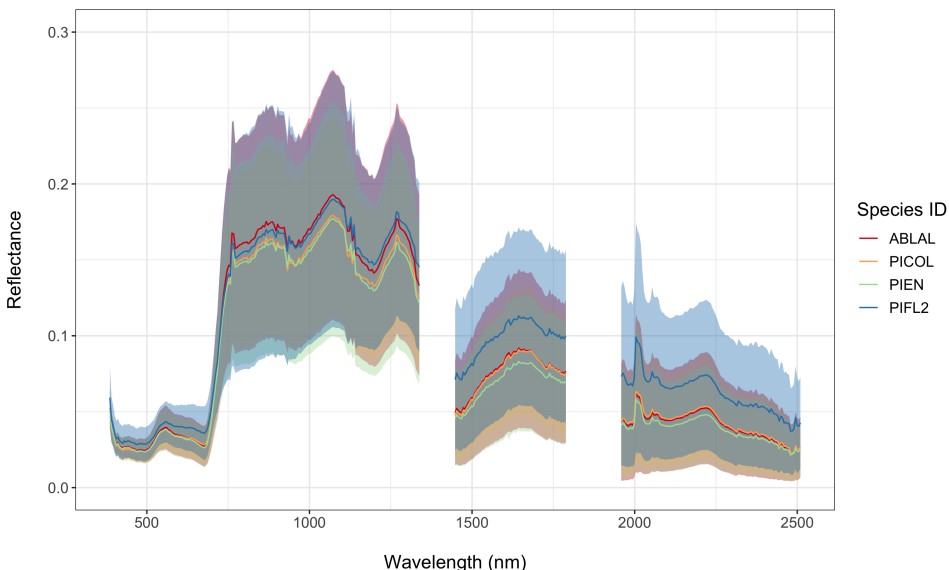

**Figure 7.** Mean hyperspectral reflectance from 380 to 2510 nm, extracted from all polygons with half the maximum crown diameter at the NEON NIWO site for each of the dominant tree species: ABLAL (Subalpine fir), PICOL (Lodgepole pine), PIEN (Engelmann spruce), and PIFL2 (Limber pine). Reflectance curves for all four species are overlaid here to compare and contrast their mean and standard deviation values. Shading illustrates +/− one standard deviation in reflectance per wavelength. Gaps in the spectra at approximately 1350 nm and 1800 nm are where "bad bands" were removed where there is high atmospheric absorption.

**Table 3.** Evaluation metrics for each random forest model using a different training data set.

| Training Set | OOB Accuracy | IV Accuracy | Kappa |
|---|---|---|---|
| Points | 0.682 | 0.458 | 0.538 |
| Polygons–half diameter | 0.690 | 0.597 | 0.573 |
| Polygons–max diameter | 0.624 | 0.590 | 0.490 |
| Points–half diam clipped | 0.598 | 0.528 | 0.434 |
| Polygons–half diam clipped | **0.693** | **0.604** | **0.578** |
| Polygons–max diam clipped | 0.645 | 0.611 | 0.516 |

Note: The evaluation metrics are: Out-of-Bag estimate of accuracy internal to the random forest classifier (OOB Accuracy), overall accuracy of the Independent Validation set predictions (IV Accuracy), and Cohen's Kappa coefficient (Kappa). The individual ID's in each of the "raw" polygon sets were reduced to match those within each of the corresponding clipped data sets. Bold text highlights the row with the highest overall accuracy based on OOB and IV accuracies.

**Table 4.** Confusion matrix with User's Accuracy (UA) and Producer's Accuracy (PA) per species for the training set that yielded the highest overall accuracy.

| | | Predicted Species | | | | |
|---|---|---|---|---|---|---|
| | | ABLAL | PICOL | PIEN | PIFL2 | PA % |
| True species | ABLAL | 28 | 18 | 31 | 9 | 32.6 |
| | PICOL | 4 | 110 | 38 | 11 | 67.5 |
| | PIEN | 8 | 30 | 111 | 19 | 66.1 |
| | PIFL2 | 1 | 2 | 5 | 149 | 94.9 |
| | UA % | 68.3 | 68.8 | 60.0 | 79.3 | |

Note: Species codes correspond to: ABLAL (Subalpine fir), PICOL (Lodgepole pine), PIEN (Engelmann spruce), and PIFL2 (Limber pine).

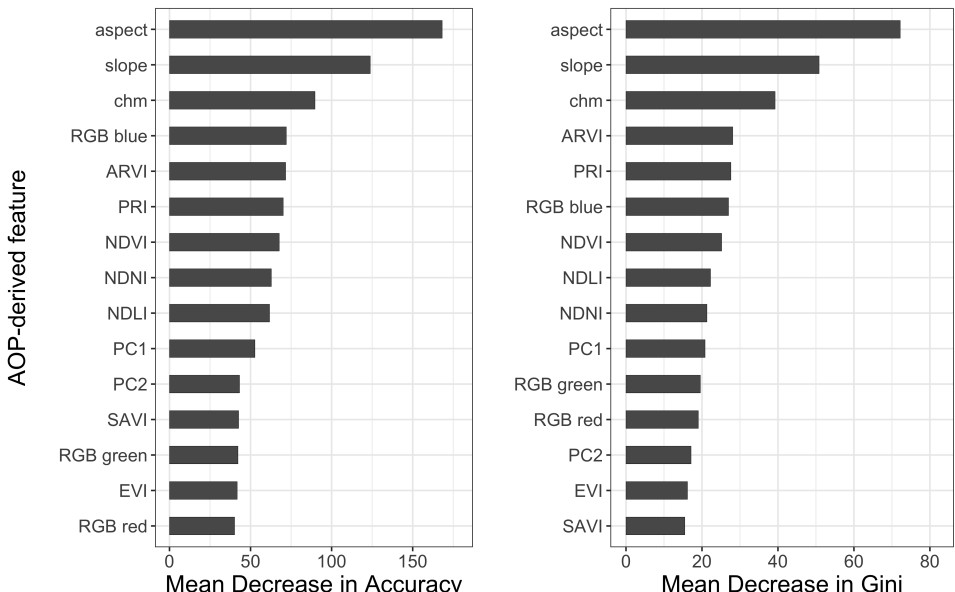

**Figure 8.** Predictor variable importance ranked by two metrics, Mean Decrease in Accuracy (MDA) and Mean Decrease in Gini Index (MDG) for the Random Forest models trained using the half-diameter clipped polygons. See Table 2 for remote sensing-derived variable definitions.

## 4. Discussion

We built an open reproducible workflow to create training data sets using NEON data without the need for manual delineation or external data sources. We represented individual trees as points and circular polygons with various sizes based on in-situ crown diameter measurements, and we implemented a clipping workflow to give preference to larger, taller trees that are more likely to be seen in layered canopies by the airborne remote sensing platform. We trained RF models to predict species using each training set paired with AOP remote sensing features, and determined which remote sensing data products were most important for species identification. Our results show that 60–69% overall classification accuracies are achievable at the NIWO site without any manual refinement of the training data or incorporation of data beyond the NEON collection protocols. These accuracy values fall within the lower end of the reported range of 65–95% for studies utilizing combined sensor systems (spectral and structural data) [6]. Based on the extracted spectra, the mean and standard deviation of reflectance for the four coniferous tree species appear to be very similar (Figure 7).

Accuracy values were highest when half-diameter circular polygons were used, rather than points or maximum-diameter polygons (Table 3). This is useful information for creating future training data sets using NEON in-situ measurements. Minimum crown diameter is also measured, and could be used to approximate circular crown size, as there are no other measurements that give a more specific indication of crown shape in the NEON data collection protocol. After applying the proposed clipping workflow, classification accuracy increased slightly (no more than 2%) between each pair of reference data sets. We expected a larger accuracy increase, under the assumption that removing tree crowns with small area values and clipping overlap between neighboring crown polygons would yield a purer spectral and structural signal for each species.

Accuracy values varied across each of the four coniferous species in our study area (Table 4). The worst classification results were obtained for Subalpine fir (ABLAL) and Engelmann spruce (PIEN), while the best classification results were obtained for limber pine (PIFL2). Based on the spectral reflectance curves in Figure 7, PIFL2 appears to be noticeably better separated from the other three species across the longer wavelengths, from 1500 nm out to 2500 m, and this spectral separation may help explain why PIFL2 was classified most accurately.

Besides the clipping workflow and using the CHM to remove pixels associated with a canopy height of zero, points and polygons derived from the in-situ tree measurements were used as-is, without any site-specific parameters, manual quality assessment, or alignment procedures. Our visualizations indicate that some stem locations and polygons extend beyond the actual location and boundaries of tree crowns in the AOP imagery (Figure 4). This could result from field measurement errors and uncertainties, georectification and image formation artifacts introduced during the creation of the remote sensing data products, and/or asymmetric crowns.

We performed pixel-based classification, but existing studies show potential for improved results using manual and automated tree crown delineation to refine the selection of pixels within their reference data sets and perform object-based classification [28,47,61]. We recognize that individual tree detection and segmentation are complicated tasks that often require site-specific tuning parameters to avoid under/over segmentation [5]. Tree detection methods often struggle with transferability across geographies and vegetation types. We aimed to create and evaluate a training data processing method that takes full advantage of standardized NEON-collected tree measurements, and avoids being site-specific. We are aware of other studies such as [62] who are making impressive strides towards creating transferable methods for large-scale tree detection across NEON sites. Note that methods like this one utilize tens of thousands of hand-annotated tree crowns for model training.

Beyond using the CHM to identify canopy pixels, additional spectral data can be used such as an NDVI threshold to isolate forest pixels and eliminate shadow pixels [36]. An NDVI threshold could also be used to identify pixels with trees that have died since the collection of the field data, as there was a time difference of 1–2 years between the in-situ and remote sensing data collections in this study. Incorporating additional steps to refine the reference data sets, ensure alignment with the remote sensing imagery, and delineate individual crown boundaries instead of assuming perfectly circular crowns may improve upon the accuracies achieved here. We anticipate that employing outlier removed steps may further separate the mean spectral reflectance curves and reduce the standard deviation of hyperspectral reflectance per species that are presented in Figures 6 and 7.

We found that the LiDAR-derived remote sensing features of aspect, slope, and canopy height were consistently ranked as the most important for species classification (Figure 8). This is interesting, as canopy height on its own is typically limited for robust species classification as it is dependent on tree age [6]. In addition to using the CHM, other studies have found that calculating metrics related to the vertical distribution, density and intensity of individual LiDAR returns improved classification results, so these metrics may be promising to incorporate in the future [6,15,63]. Incorporating LiDAR point cloud-based metrics per pixel and/or tree crown object may improve species classifier performance in the future. Aspect and slope are important drivers of microclimate conditions such as temperature, moisture, and sun and wind exposure, so their importance has ecological merit in the mountainous landscape of the Southern Rockies. The importance of aspect and slope in our variable assessment may indicate that this classifier is not just learning how to identify trees using inter-species spectral variability, but also incorporating spectral information from the habitat or niche where each species resides. Future work would be required to assess the consistency of these observations with documented patterns of species distributions, but this highlights an interesting potential avenue for assessing how species habitats may shift in mountainous regions across the 30 year duration of the NEON project.

Following the LiDAR-derived variables, the ranking of RGB and NIS-derived variable importance varied slightly between importance metrics (Figure 8). These variable importance rankings may be useful for iterative variable selection in future classification efforts, such as backwards feature selection [36]. In addition, performing a sensitivity analysis would be valuable to fine-tune the *ntree* and *mtry* RF parameters in subsequent analyses to potentially achieve higher species classification accuracies.

From the AOP hyperspectral imagery, we utilized a series of vegetation indices and the first two principal components (PCs) in our RF models. To further improve the utility of the hyperspectral

data, we consider employing other dimensionality reduction techniques and including additional features in future work. We incorporated the first two PCs, which explained 96% of the hyperspectral data. Including additional PCs to explain additional variance may improve our classification results in future work. PCA and Minimum Noise Fraction (MNF) were the most commonly used dimensionality reduction methods in a recent tree species classification review [6]. There is also potential for MNF to improve classification results, and this would be an interesting comparison to make in future research. Hyper-dimensional image data volume may also be reduced by selecting specific spectral bands that are shown to successfully discriminate species [15]. There are specific wavelengths that have been shown to vary as a function of vegetation type and health, the foundation for vegetation indices as well, such as reflectance at 550 nm (the green peak within the visible wavelengths) [64] and reflectance at 750 nm (at the NIR shoulder) [65]. Additionally, spectral separability analysis could be useful to analyze our extracted reflectance curves presented in Figures 6 and 7. Separability metrics such as the Jeffries–Matusita distance and transformed divergence have been used to identify wavelengths that play a significant role in vegetation classification [66]. Regarding the contribution of the RGB imagery, we consider using texture metrics in future analyses to quantify spatial patterns within tree crowns such as shadows and other foliage characteristics to potentially improve classification results [35]. Calculating explicit texture metrics may also help to ensure that the RGB data is not adding redundant information that is already described by the visible bands within the hyperspectral data.

This analysis can also be performed at other NEON sites to generate additional training data sets for regional species classification. However, the types of vegetation and topography vary among NEON domains, and we expect this to influence the resulting variable importance rankings and species classification accuracies. For instance, the LiDAR-derived features of slope and aspect describe the underlying terrain steepness and orientation. These two variables were found to be important for species classification at the mountainous NIWO site in the Southern Rockies in our study. However, the Ordway–Swisher Biological Station (OSBS) site in north-central Florida is relatively flat. We do not expect the slope and aspect variables to be as useful for species classification at a site such as OSBS where slope and aspect are relatively constant across space.

We expect overall species classification to be influenced by the diversity and tree canopy complexity at different ecosystems across NEON sites. For instance, San Joaquin Experimental Range (SJER) in central California features open woodland dominated by large oak trees, pine trees, and scattered shrubs and grasses. Harvard Forest (HARV) in Massachusetts features primarily closed-canopy mixed forest composed of both coniferous and hardwood trees, densely packed with overlapping crowns. We expect greater species classification accuracies to be achieved at SJER compared to at HARV, because an open woodland offers simpler canopy structure and clearer separation between neighboring tree crowns. In addition, open woodland enables more accurate GPS measurements to be collected at plot corner locations as opposed to in dense, closed-canopy tree cover.

Incorporating additional training data and/or implementing a more complex classification model may improve our results. Complete field data entries for just 699 trees were available at the NIWO site for our analysis. Deep learning methods are gaining traction in the forest remote sensing field for their ability to perform complicated tasks such as tree detection and species classification, although they require large volumes of training data [32]. Deep learning methods may also enable more effective transferability of species classification methods across geographies and vegetation types. When a recently developed deep learning classification method was applied across multiple NEON sites, tree detection was the least successful at the NIWO site compared to three other NEON sites with oak woodland, mixed pine, and dense deciduous forest types [62]. Taking both our results and the findings of [62] into consideration, species classification may objectively be a difficult task at the NIWO site due to the spectral similarity and structural characteristics of the coniferous vegetation species present there.

## 5. Conclusions

In this study, we created a series of training data sets using NEON in-situ vegetation measurements and used each one to train a random forest model to classify tree species at the NIWO subalpine site. The highest classification accuracies, 69% and 60% based on internal out-of-bag error and an independent validation set, respectively, were achieved using circular polygons created with half the maximum crown diameter per tree after applying the proposed clipping workflow. The LiDAR-derived raster data products of aspect, slope, and canopy height were found to be the most important AOP remote sensing data-derived features, followed by vegetation indices. Refining the reference data alignment with remote sensing imagery, incorporating additional variables from the LiDAR point cloud, and performing object-based classification on crown segments, rather than individual pixels, are all promising directions for subsequent analyses at NIWO and in subalpine coniferous forests more generally. Scaling this species classification method to other NEON sites will also be informative of the proposed workflow's robustness. This work contributes to the open development of well-labeled training data sets for forest composition mapping using openly available NEON data without requiring external data collection, manual delineation steps, or site-specific parameters.

**Author Contributions:** V.M.S., M.E.C., and J.K.B. conceived of project design. All authors collaboratively developed and refined the methodology. V.M.S. performed the analysis and wrote the text. All authors contributed to reviewing and editing the text. All authors have read and agreed to the published version of the manuscript.

**Funding:** Funding for this work was provided by Earth Lab, through CU Boulder's Grand Challenge Initiative, and the CIRES at CU Boulder. Additional funding was provided by National Aeronautics and Space Administration (NASA) New Investigator Program (NIP) grant 80NSSC18K0750 to M. E. Cattau.

**Acknowledgments:** The National Ecological Observatory Network is a program sponsored by the National Science Foundation and operated under cooperative agreement by Battelle Memorial Institute. This material is based in part upon work supported by the National Science Foundation through the NEON Program.

**Conflicts of Interest:** The authors declare no conflict of interest.

## Abbreviations

The following abbreviations are used in this manuscript:

| | |
|---|---|
| NEON | National Ecological Observatory Network |
| AOP | NEON's Airborne Observation Platform |
| NIWO | Niwot Ridge Mountain Research Station NEON site |
| LiDAR | Light Detection and Ranging |
| CHM | Canopy Height Model |
| NIS | NEON Imaging Spectrometer |
| RGB | Red, Green, Blue multispectral imagery |
| RF | Random Forest classification |

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
