# Peer review of "Integrating National Ecological Observatory Network (NEON) Airborne Remote Sensing and In-Situ Data for Optimal Tree Species Classification"

_remotesensing, doi:10.3390/rs12091414_

Round 1

Reviewer 1 Report

Title: NEON – the instructions for authors state that the abbreviations should be used for gene or protein names… Maybe better to use National Ecological Observatory Network? Anyway, why the airborne remote sensing gets specific due to NEON? What is optimal?

Abstract: What is AOP (it is defined in page 4, that you are dealing with a specific airborne observation platform)? Underline the “novelty” of your approaches. Generally, the abstract provides information sufficient to understand your approaches and findings only for the reader, who is familiar with NEON. However, the message for more generalist reader could be stronger formulated.

Introduction

The introduction should be improved, first of all bearing in mind, that the manuscript is submitted to remote sensing journal. I find paragraph (lines 20-26) redundant – it looks too weak for remote sensing journal, there is no need to say for remote sensing community that the remote sensing is OK. I suppose you can easily start with the NEON and its role in remote sensing and ecological studies and applications. More information is needed on the use of NEON data for research. You guide the reader to several references to search for information, however, it is not clear what is the status of the research field, what are the knowledge gaps and your niche. Why do you need your study? What will be your research contribution? Formally, the journal also requires to formulate the hypotheses and the aims of your study. Finally, your statement on “initial assessments” (line 52) may lead to the question whether your report is not still premature for RS.

Methods:

Line 97: you state that you used available point cloud-derived raster data products. LiDAR point cloud data is available from NEON – so how can you be sure that standardised data set provides the best possible inputs for your tests? Having original point clouds available researcher could always go above the limits of some standardized data set, simultaneously detecting such limits.

Figure 3 – why do you use true-colour composites for advanced HSI image – I suppose you will find it as more advantageous to RGB image in context of tree species identification?

Line 131: what is “shorter polygon”?

Reference data: why did you focus just on geo-processing solutions to get the reference information for crowns? Having information on tree locations and crown parameters you could easily conduct more sophisticated study to detect the pixels referring to each specific crown - segmentation, finally, manual interpretation to get some validation data set to test your approach. OK, you have got six reference data sets, but, hardly anyone may be considered as the best describing the trees under focus.

What is the pixel size of NIS? You did not apply any resampling on HSI data? How do you motivate the RGB resampling into 1x1 m raster?

Lines 180-195: not sure whether you need detailed description of RF algorithm – it is very well know and widely used in similar studies. Why did you choose RF? Or just RF? Why did you not consider some more sophisticated algorithms, assuming that may researchers nowadays pretend to be innovative with “deep learning” etc. (even though the data sets are small)

Results:

Fig 6 looks redundant. Why you did not attempt to analyse the reflectance curves? You could get the potential to discriminate the species based on reflectance data before starting the manipulations with RF. There are numerous studies processing HSI data published during last decade, to look for the methods.

Discussion:

The major issue of your study seems to be that you consider your methodological approach (e.g. to get the reference data) as the only one. It is unfair to compare just the conclusive results without considering the conditions of other studies. Say, that your approach to treat the reference data is already a routines procedure in NEON, then there is no need for study. The major issue is how to associate the image pixel with a specific tree and there are other potential methodological solutions you have not considered.

Assuming, that results are what they are, some findings need to be discussed. E.g. how do you explain relatively largest importance of terrain aspect and slope, is this not related to your approach to create the reference data. How do you explain the errors in identification of specific tree species, maybe there were other factors important to consider, like the crown size = age = different spectral features within the same species, etc. How do you propose using your findings outside the NEON sites?

Other:

SI Units (International System of Units) should be prioritized following the requirements for authors (e.g. Fig. 2).

Reviewer 2 Report

The manuscript submitted for the review concerns an important issue - the preparation of reference data for the tree species classification using multisensor remote sensing data. The Authors use open ecological data from the National Ecological Observatory Network (NEON) for this purpose. It is very interesant initiative.
In terms of clarity, the terminology used and the preparation of drawings, the article should be assessed well. In terms of clarity, the terminology used and the preparation of drawings, the article should be assessed well. Unfortunately, the concept of research and description of the results obtained definitely require significant improvements. Preparing a reference (training data set) in a way suggested by the Authors is nothing new. Most scientists use buffers around points as the easiest way to generate reference datsets for the classification. However, in addition, they use a much broader spectrum of methods to reduce or eliminate the impact of shadows, neighborhood, etc. From this point of view, it is difficult to consider the article as significantly innovative. The fact that the obtained accuracy is relatively low may be due to the way the reference is prepared.

General comments:

  • The literature review is quite cursory. The introduction is a very short description, almost without reviewing the status of existing studies. The text of the article notes a lack of good recognition of very rich literature in this specific topic. Only few references refer to the issue discussed.
  • Authors should test more widely the methods of generating reference polygons - this is a key element affecting the achievement of correct classification results, so eliminating, for example, parts of buffers where there are no trees, where there are shadows, etc. is the success key.
  • The selection of features for classification was carried out arbitrarily. This was partly justified, but not entirely convincing. For example, the use of PCA components usually gives worse results in the classification based on hyperspectral data than the use of MNF components, so this approach seems more appropriate. In addition, the authors considered only the first 2 principal components that may have affected the weaker classification results, as some species-differentiating features may be included in the further principal components. In turn, adding RGB data in the manner proposed by the authors seems to partially duplicate information about the brightness in the RGB spectrum.
  • The worst classification results were obtained for the species that was the least numerous in the reference data set, which may be the reason for such a result. When analyzing the average spectral characteristics, it seemed that there should be no big problem distinguishing them. It happened differently. Why? Maybe the reference set for this species contained the most errors, the location of the measured points was shifted? It is worth looking at this issue a bit better.
  • There should be more information in the manuscript about the results obtained from other training sets. Are confusion matrices for other results analogous to those obtained for the best classification?

Detailed comments:

  • Line 169: I cannot agree with the statement that: "The slope and aspect products describe the orientation and steepness of the underlying" bare earth "surface, which are important microclimate characteristics that influence species survival within a given environment [23]" . If the Authors calculated slope and aspect based on LiDAR-derived CHM, then these parameters characterize the structure of the tree crowns, not the terrain surface.

  • Line 205-207: reference to literature needed.

  • Line 312-314: As already mentioned, the slope and aspect calculated on the basis of CHM characterize the vegetation canopy, not the terrain surface.

I recommend, among others the following references:

  1. Ballanti, L.; Blesius, L.; Hines, E.; Kruse, B. Tree species classification using hyperspectral imagery: A comparison of two classifiers. Remote Sens. 2016, 8, 445.
  2. OsiÅ„ska-Skotak, K.; Radecka, A.; Piórkowski, H.; Michalska-Hejduk, D.; Kopeć, D.; Tokarska-Guzik, B.; Ostrowski, W.; Kania, A.; Niedzielko, J. Mapping Succession in Non-Forest Habitats by Means of Remote Sensing: Is the Data Acquisition Time Critical for Species Discrimination? Remote Sens. 2019, 11, 2629.

  3. Piiroinen, R.; Heiskanen, J.; Maeda, E.; Viinikka, A.; Pellikka, P. Classification of tree species in a diverse African agroforestry landscape using imaging spectroscopy and laser scanning. Remote Sens. 2017, 9, 875.
  4. Wietecha, M.; Jełowicki, M.; Mitelsztedt, K.; Miścicki, S.; Stereńczak, K. The capability of species-related forest stand characteristics determination with the use of hyperspectral data. Remote Sens. Environ. 2019, 231, 111232. doi: 10.1016/j.rse.2019.111232.

  5. Hauglin, M.; Ørka, H.O. Discriminating between native Norway spruce and invasive Sitka spruce—A comparison of multitemporal Landsat 8 imagery, aerial images and airborne laser scanner data. Remote Sens. 2016, 8, 363.

  6. Ghiyamat, A.; Shafri, H.Z.M.; Shariff, A.R.M. Influence of tree species complexity on discrimination performance of vegetation indices. J. Remote Sens. 2016, 49 (1), 15–37. doi: 10.5721/EuJRS20164902

Reviewer 3 Report

The paper presents a classification scheme for tree species classification based on random forest in NEON database. The paper is written well in a structure that is easy for a reader to follow. The results presented in the article are interesting. The scientific tone of the article is adjust and acceptable. The English language is very well and acceptable. The approach has some flaw that needs to be generally addressed by a future work. In addition to the approach that is presented by this article, a good description about NEON is presented.

In my opinion, this article is eligible to be published after minor revision. Here, I list my recommendations, however, I don’t need all of my comments to be addressed on the final version:

  • Line 12: “Polygons” needs to be replaced with circle, since no real boundary polygon has been used in the analysis and this could be misleading.
  • The number of tree species that are classified in this work needs to be stated in the abstract (I guess it is four).
  • The contribution of this work needs to be highlighted and improved in the final sentence of the abstract.
  • Line 21: “Airplanes”-> “aerial vehicles” or “Airplanes and UAVs”
  • “object-based image analysis”-> objects or regions that are extracted by image analysis.
  • Line 34: “multispectral and hyperspectral images and LIDAR data can be all used”
  • Line 35: I recommend referring also to the following papers:

Yu, X.; Hyyppä, J.; Litkey, P.; Kaartinen, H.; Vastaranta, M.; Holopainen, M. Single-Sensor Solution to Tree Species Classification Using Multispectral Airborne Laser Scanning

Nezami, Somayeh, et al. "Tree Species Classification of Drone Hyperspectral and RGB Imagery with Deep Learning Convolutional Neural Networks." Remote Sensing 12.7 (2020): 1070.

  • Line 36 seems a bit confusing to me: Does it mean that at the same time when AOP data is collected (which is a continuous data acquisition because present verb is used), in-site measurements of individual trees are observed? If yes, then please consider rewriting this sentence.
  • Line 45 and 46: The aim of combining these two data (NEON and AOP in NIWO) is stated “to evaluate the impact of different data fusion approaches for generating species classification training data.” This aim is a bit vague in my opinion, and could be improved by saying “to evaluate the impact of different data fusion on tree species classification accuracy.”

  • The introduction lacks a literature review on recent tree species classification. I recommend adding at-least a paragraph to review state-of-the-art techniques.
  • At the last part of the introduction, a comparison to recent state-of-the-art classification techniques is required to make the contributions of this article more visible.
  • I recommend to state the contributions of this work in as a list of bulleted items.
  • Line 71: 30cm accuracy for how long distance?
  • Line 79: You can maybe list those three remote-sensing instruments here.
  • Flying altitude, ground sampling distance (GSD), and Field of View (FOV) could be stated in 2.2
  • Line 91: Those seven indices could be briefly listed here.
  • Line 97 “the readily available point cloud-derived raster data products” where does “the” refer to (does it previously mentioned)?
  • Line 99: “ and a surface of normalized…”
  • Line 105: Spatial resolution of 10 cm for which flight altitude?
  • Line 112: Is the crown diameters measured around each tree or of a single measurement is used for each tree?
  • Line 112: “crown diameters”->” crown diameter”
  • Line 151 add a hyphenation between water and vapor “Water-vapor absorption bands”
  • Line 156, 157: It is a big concern to me that you used 96 % of total variance. One reason for low classification accuracy could be this step. It is possible that you loose valuable feature in that 4%. Moreover, the dimensionality problem could be addressed by employing other classification technique. You can consider this comment for your future development. For this article, you may write a sentence stating that PCA with 4% threshold could be a possible factor for relatively low classification accuracy.
  • Table 3, OOB should be described in the Table description. The table is supposed to be self-descriptive.
  • Line 255: Based on recent literature, deep-learning is one of the best classifiers to deal with high-dimensional data along with other non-linear classifiers such as multi-layer perceptron fully connected model. A single article indicating a low accuracy for deep learning could not be considered as sufficient proof that deep learning doesn’t work well. Other reasons such as a lack of an accurate radiometric adjustment, employing PCA with cutting important variance, low-quality CHM, and using RF could be other reasons for relatively low classification accuracy.
  • I recommend adding few sentences to describe the possible reason that this classification accuracy is less that the state-of-the-art research works.

And as a last note, I recommend the authors to deeply consider radiometric adjustment and machine learning aspects in their future publication. Good Luck!

Round 2

Reviewer 1 Report

I accept the modifications done by the authors and thank for the exhaustive comments on the changes done.

Reviewer 2 Report

Dear Authors,

Thank you for considering my suggestions and providing comprehensive answers to my questions and comments. I believe that the extension of some elements proposed by all reviewers significantly increases the scientific value of the paper.

Good luck in your further research work!